# SceneScape: Text-Driven Consistent Scene Generation

**Rafail Fridman**[*]
Weizmann Institute of Science
rafail.fridman@weizmann.ac.il

**Amit Abecasis**[*]
Weizmann Institute of Science
amit.abecasis@weizmann.ac.il

**Yoni Kasten**
NVIDIA Research
ykasten@nvidia.com

**Tali Dekel**
Weizmann Institute of Science
tali.dekel@weizmann.ac.il

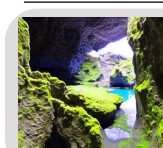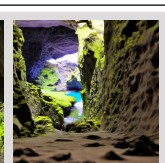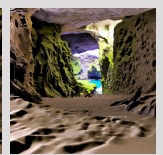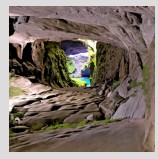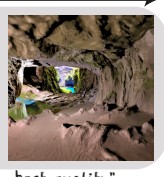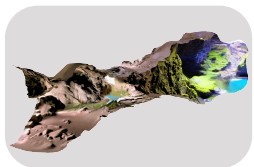

"POV, cave, pools, water, dark cavern, inside a cave, beautiful scenery, best quality"

**(a) Sample frames from our generated video**

**(b) Our unified representation**

Figure 1: (a) Given a text prompt describing a scene, *SceneScape* generates a long-term video depicting a walkthrough of the scene, adhering to a given camera trajectory. We achieve this through a test-time optimization approach that utilizes a pre-trained text-to-image generative model and a depth prediction model, enabling online 3D-consistent perpetual view generation without the need for training data. (b) To synthesize a 3D-consistent video , we construct a unified 3D mesh concurrently with the video. For visualization purposes, we remove the side part of the mesh.

## Abstract

We present a method for *text-driven perpetual view generation* – synthesizing long-term videos of various scenes solely from an input text prompt describing the scene and camera poses. We introduce a novel framework that generates such videos in an online fashion by combining the generative power of a pre-trained text-to-image model with the geometric priors learned by a pre-trained monocular depth prediction model. To tackle the pivotal challenge of achieving 3D consistency, i.e., synthesizing videos that depict geometrically-plausible scenes, we deploy an online test-time training to encourage the predicted depth map of the current frame to be geometrically consistent with the synthesized scene. The depth maps are used to construct a *unified* mesh representation of the scene, which is progressively constructed along the video generation process. In contrast to previous works, which are applicable only to limited domains, our method generates diverse scenes, such as walkthroughs in spaceships, caves, or ice castles. Project page: *https://scenescape.github.io/*

## 1 Introduction

By observing only a single photo of a scene, we can imagine what it would be like to explore it, move around, or turn our head and look around – we can envision the 3D world captured in the image and expand it in our mind. However, achieving this capability computationally – generating a long-term video that depicts a plausible visual world from an input image – poses several key challenges. One of the main challenges is ensuring that the synthesized content is consistent with a feasible 3D world. For example, it must correctly depict parallax effects and occlusion relations of different objects in

---

[*]Equal contribution

37th Conference on Neural Information Processing Systems (NeurIPS 2023).

the scene. Additionally, to synthesize new content, a strong prior about the visual world and how it would appear beyond the current field of view is necessary. Finally, the generated content should appear smooth and consistent over time.

Previous methods have tackled the task of perpetual view generation for specific domains, for example, synthesizing landscape flythroughs [e.g., 5, 7, 28, 29], or bedrooms walkthroughs [e.g., 45, 47]. These methods involve large-scale training on videos or images from the target domain, which limits their use. In this work, inspired by the transformative progress in text-to-image generation, we take a different route and propose a new framework for *text-driven perpetual view generation* – synthesizing long-range videos of scenes solely from free-vocabulary text describing the scene and camera poses. Our method does not require any training data but rather generates the scene in a zero-shot manner by harnessing the generative prior learned by a pre-trained text-to-image diffusion model and the geometric priors learned by a pre-trained depth prediction model.

More specifically, given an input text prompt and a camera trajectory, our framework generates the video in an online fashion, one frame at a time. The text-to-image diffusion model is used to synthesize new content revealed as the camera moves, whereas the depth prediction model allows us to estimate the geometry of the newly generated content. To ensure that the generated scene adheres to a feasible 3D geometry, our framework estimates a unified mesh representation of the scene that is constructed progressively as the camera moves in the scene. Since monocular depth predictions tend to be inconsistent across different frames, we finetune the depth prediction model at test time to match the projected depth from our mesh representation for the known, previously synthesized content. This results in a test-time training approach that enables diverse, 3D-consistent perpetual view generation.

To summarize, our key contributions are the following:

1. The first text-driven perpetual view generation method, which can synthesize long-term videos of diverse domains solely from text and a camera trajectory.

2. The first zero-shot/test-time 3D-consistent perpetual view generation method, which synthesizes diverse scenes without large-scale training on a specific target domain.

3. Achieving 3D-consistent generation by progressively estimating a unified 3D representation of the scene.

We thoroughly evaluate and ablate our method, demonstrating a significant improvement in quality and 3D consistency over existing methods.

## 2 Related Work

**Perpetual view generation.** Exploring an "infinite scene" dates back to the seminal work of Kaneva et al. [24], which proposed an image retrieval-based method to create a 2D landscape that follows a desired camera path. This task has been modernized by various works. Different methods proposed to synthesize scenes given as input a single image and camera motion [e.g., 21, 26, 28, 29, 45, 46, 56, 58, 63]. For example, synthesizing indoor scenes [26, 45, 46, 58], or long-range landscapes flythroughs [28, 29]. These methods have demonstrated impressive results and creative use of data and self-supervisory signals [e.g., 5, 7, 11, 12, 28, 45]. However, none of these methods considered the task of *text-driven* perceptual view generation and rather tailored a model for a specific *scene domain*. Thus, these methods require extensive training on a large-scale dataset to learn proper generative and geometric priors of a specific target scene domain.

**3D-Consistent view synthesis from a single image.** Various methods have considered the task of novel view synthesis from a single input image. A surge of methods has proposed to utilize a NeRF [35] as a unified 3D representation while training on a large dataset from a specific domain to learn a generic scene prior [22, 60, 62]. Other 3D representations have been used by various methods, including multi-plane images [57, 66], point clouds which hold high-dimensional features [46, 58], or textured-mesh representation [20]. In Jain et al. [22], the task is further regularized using the semantic prior learned by CLIP [41], encouraging high similarity between the input image and each novel view in CLIP image embedding space. All these methods require training a model on specific domains and cannot produce perpetual exploration of diverse scenes.

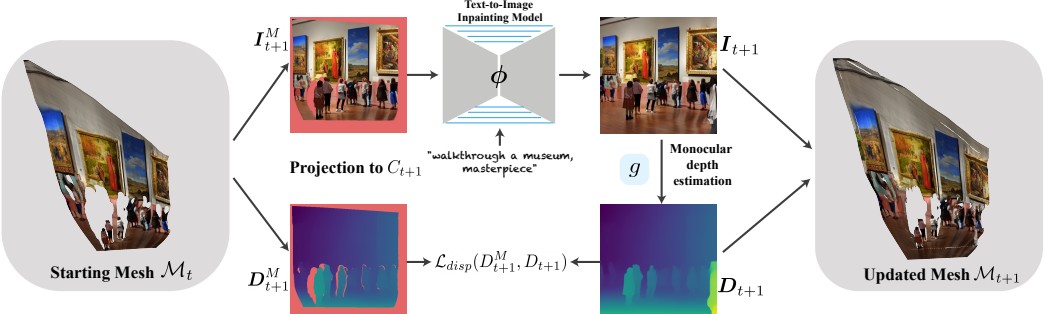

Figure 2: **SceneScape pipeline.** We represent the generated scene with a unified mesh $\mathcal{M}$, which is constructed in an online fashion. Given a camera at $C_{t+1}$, at each synthesis step, a new frame is generated by projecting $\mathcal{M}_t$ into $C_{t+1}$, and synthesizing the newly revealed content by using a pre-trained text-to-image diffusion model. To estimate the geometry of the new synthesized content, we leverage a pre-trained depth prediction model; to ensure the predicted depth is consistent with the existing scene $\mathcal{M}_t$, we deploy a test-time training, encouraging the predicted depth by the model to match the projected depth from $\mathcal{M}_t$. We then update our mesh representation to form $\mathcal{M}_{t+1}$ which includes the new scene content.

**3D-Aware image generation.** Related to our task is 3D-aware image synthesize, which typically involves incorporating a 3D representation explicitly into a generative model, e.g., a NeRF representation [8, 9, 16, 36, 50], or neural surface renderer [38]. These methods have shown impressive results on structured domains (e.g., cars or faces), but deploying them for diverse scenes is extremely challenging and often requires additional guidance [13], or heavy supervision, e.g., large-scale data that contains accurate camera trajectories, which can be obtained reliably mainly for synthetic data [4]. We aim for a much more diverse, flexible, and lightweight generation of arbitrary scenes.

**Text-to-video generation and editing.** Our work follows a broad line of works in the field of text-to-video synthesis and editing. There has been great progress in developing text-to-video generative models by expanding architectures to the temporal domain and learning priors from large-scale video datasets [e.g. 15, 18, 32, 51]. Nevertheless, these models are still in infancy, and are lagging behind image models in terms of quality (resolution and video length). On the other side of the spectrum, a surge of methods proposes to tune or directly leverage a 2D text-image model for video editing tasks [e.g., 3, 6, 40, 59, 65]. We also follow this approach in conjunction with generating an explicit 3D scene representation.

**Text-to-3D Generation and Editing.** Recently, language-vision pre-trained models [41, 49], and differentiable renderers [30, 35] have been utilized for text-driven 3D content generation. For example, CLIP-based losses have been used for the text-driven stylization of a given surface mesh [34] or for synthesizing a NeRF from an input text [23]. Taking this approach further, Poole et al. [39] distills the generative prior of a pre-trained text-to-image diffusion model by encouraging the optimized NeRF to render images that are in the distribution of the diffusion model [49]. Singer et al. [52] extends this approach to a 4D dynamic NERF and optimizes using a pre-trained text-to-video model. Nevertheless, it is limited to object-centric scenes while we generate long-term walkthrough-type videos. On the other hand, Chang et al. [10] and Ma et al. [33] construct the semantic scene graph from natural language and retrieve the required elements from the 3D objects databases.

Finally, in a very recent concurrent work, Text2Room [19] proposed a similar approach to ours to generate 3D scenes based on textual prompts. However, this work focuses specifically on creating *room meshes* and tailors both the camera trajectory, prompts and mesh rendering for this purpose. In contrast, we use the mesh representation only as a tool to achieve *videos* of diverse scenes.

## 3 Method

The input to our method is a text prompt $P$, describing the target scene, and a camera trajectory denoted by $\{C_i\}_{i=1}^{T}$, where $C_i \in \mathbb{R}^{3 \times 4}$ is the camera pose of the $i^{th}$ frame. Our goal is to synthesize a long-term, high-quality video depicting the desired scene while adhering to the camera trajectory.

Our framework, illustrated in Fig. 2, generates the video one frame at a time, by leveraging a pre-trained text-to-image diffusion model [48], and a pre-trained monocular depth prediction model [42, 43]. The role of the diffusion model is to synthesize new content that is revealed as the camera moves, whereas the depth prediction model allows us to estimate the geometry of the newly generated content. We combine these two models through a unified 3D representation of the scene, which ensures geometrically consistent video generation. More specifically, at each time step $t$, the new synthesized content and its predicted depths are used to update a unified mesh representation $\mathcal{M}$, which is constructed progressively as the camera moves. The mesh is then projected to the next view $C_{t+1}$, and the process is repeated.

Since neither the depth prediction model nor the inpainting model are guaranteed to produce consistent predictions across time, we finetune them both to match the content of our unified scene representation for the known, previously synthesized content. We next describe each of these components in detail.

## 3.1  Initialization and Scene Representation

We denote the pre-trained text-to-image inpainting model by $\phi$, which takes as input a text prompt $P$, a mask $M$, and a masked conditioning image $I_t^M$:

$$I_t = \phi(M, I_t^M, P) \tag{1}$$

The first frame in our video $I_0$ is generated by sampling an image from $\phi$, using $P$ without masking (all-ones mask). The generated frame is then fed to a pre-trained depth prediction model $g$, which estimates $D_0$ the depth map for $I_0$, i.e., $D_0 = g(I_0)$.

With the estimated depth and camera poses, a naïve approach for rendering $I_{t+1}$ is to warp $I_t$ to $C_{t+1}$, e.g., using 2D splatting [37]. However, even with consistent depths, rendering a 3D-consistent scene is challenging. In typical camera motion, multiple pixels from one frame are mapped into the same pixel in the next frame. Thus, the aggregated color needs to be selected carefully. Furthermore, frame-to-frame warping lacks a unified representation, thus, once the scene content gets occluded, it will be generated from scratch when it gets exposed again

To address these challenges, we take a different approach and represent the scene with a unified triangle mesh $\mathcal{M} = (V, F, E)$ which is represented by a tuple of vertices (3D location and color), faces and edges, respectively. We initialize $\mathcal{M}_0$ by unprojecting $(I_0, D_0)$. After the first frame is generated, each synthesis step consists of the following main stages:

## 3.2  Project and Inpaint

The current scene content, represented by the unified mesh representation $\mathcal{M}_t$, is projected to the next camera $C_{t+1}$:

$$(I_{t+1}^M, D_{t+1}^M, M_{t+1}) = Proj\left(\mathcal{M}_t, C_{t+1}\right) \tag{2}$$

producing a mask $M_{t+1}$ of the visible content in $C_{t+1}$, a masked frame $I_{t+1}^M$, and a masked depth map $D_{t+1}^M$. Given $M_{t+1}$ and $I_{t+1}^M$, we now turn to the task of synthesizing new content. To do so, we apply $\phi$ using Eq. 1.

## 3.3  Test-time Depth Finetuning

We would like to estimate the geometry of the new synthesized content and use it to update our unified scene mesh. A naïve approach to do so is to estimate the depth directly from the inpainted frame, i.e., $D_{t+1} = g(I_{t+1})$. However, monocular depth predictions tend to be inconsistent, even across nearby video frames [27, 31, 64]. That is, there is no guarantee the predicted depth $D_{t+1}$ would be well aligned with the current scene geometry $\mathcal{M}_t$. We mitigate this problem by taking a test-time training approach to finetune the depth prediction model to be as consistent as possible with the current scene geometry. This approach is inspired by previous methods [27, 31, 64], which encourage pairwise consistency w.r.t optical flow observations in a given video. In contrast, we encourage consistency between the predicted depth and our unified 3D representation in the visible regions in frame $t + 1$. Formally, our training loss is given by:

$$\mathcal{L}_{disp} = \left\| D_{t+1}^M - g(I_{t+1}) \odot M_{t+1} \right\|_1 \tag{3}$$

where $D_{t+1}^M$, $M_{t+1}$ are given by projecting the mesh geometry to the current view (Eq. 2). To avoid catastrophic forgetting of the original prior learned by the depth model, we revert $g$ to its original weights at each generation step.

### 3.4 Test-time Decoder Finetuning

Our framework uses Latent Diffusion inpainting model [48], in which the input image and mask are first encoded into a latent space $z = E_{\text{LDM}}(M_t, I_t^M)$; the inpainting operation is then carried out in the latent space, producing $\hat{z}$, and the final output image is given by decoding $\hat{z}$ to an RGB image: $D_{\text{LDM}}(\hat{z})$. This encoding/decoding operation is lossy, and the encoded image is not reconstructed exactly in the unmasked regions. Following Avrahami et al. [2], for each frame, we finetune the decoder as follows:

$$\mathcal{L}_{dec} = \left\| D_{\text{LDM}}(\hat{z}) \odot M_{t+1} - I_{t+1}^M \right\|_2 + \left\| (D_{\text{LDM}}(\hat{z}) - D_{\text{LDM}}^{\text{fixed}}(\hat{z})) \odot (1 - M_{t+1}) \right\|_2 \quad (4)$$

The first term encourages the decoder to preserve the known content, and the second term is a prior preservation loss [2]. As in Sec. 3.3, we revert the weights of the decoder to its original weights.

### 3.5 Online Mesh Update

Given the current mesh $\mathcal{M}_t$, and the inpainted frame $I_{t+1}$, we use our depth map and camera pose to update the scene with the newly synthesized content. Note that in order to retain the previously synthesized content, some of which may be occluded in the current view, we consider only the new content, unproject, and mesh it, adding it to $\mathcal{M}_t$. That is,

$$\tilde{\mathcal{M}}_{t+1} = UnProj\left(M_{t+1}, I_{t+1}, D_{t+1}, C_{t+1}\right) \quad (5)$$

Finally, $\tilde{\mathcal{M}}_{t+1}$ is merged into the mesh:

$$\mathcal{M}_{t+1} = \mathcal{M}_t \cup \tilde{\mathcal{M}}_{t+1} = \left(V_t \cup \tilde{V}_{t+1}, F_t \cup \tilde{F}_{t+1}, E_t \cup \tilde{E}_{t+1}\right) \quad (6)$$

In practice, we also add to the mesh the triangles used to connect the previous content to the newly synthesized one. See Sec. A.1 of appendix for more details.

### 3.6 Rendering

Rendering a high-quality and temporally consistent video from our unified mesh requires careful handling. First, stretched triangles may often be formed along depth discontinuities. These triangles encapsulate content that is unseen in the current view, which we would like to inpaint when it becomes visible in other views. To do so, we automatically detect stretched triangles based on their normals and remove them from the updated mesh.

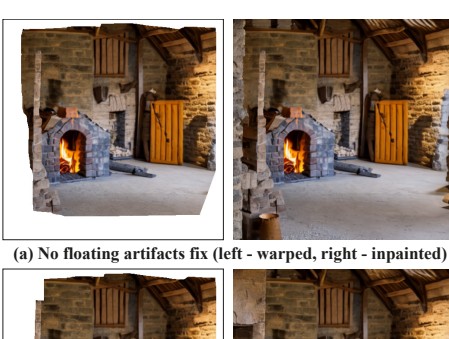

(a) No floating artifacts fix (left - warped, right - inpainted)

Another issue arises when the content at the border of the current frame is close to the camera relative to the scene geometry. This content is typically mapped towards the interior of the next frame due to parallax. This often results in undesirable "floating artifacts" where the revealed border regions are rendered with the existing distant content (see Fig. 3). We assume that such regions should be inpainted, thus we automatically detect and include them in the inpainting mask. See Sec. A.2 of appendix for more details.

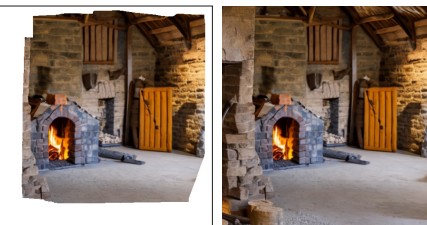

(b) With floating artifacts fix (left - warped, right - inpainted)

Figure 3: **Handling floating artifacts**

## 4 Results

We evaluate our method both quantitatively and qualitatively on various scenes generated from a diverse set of prompts, including photorealistic as well

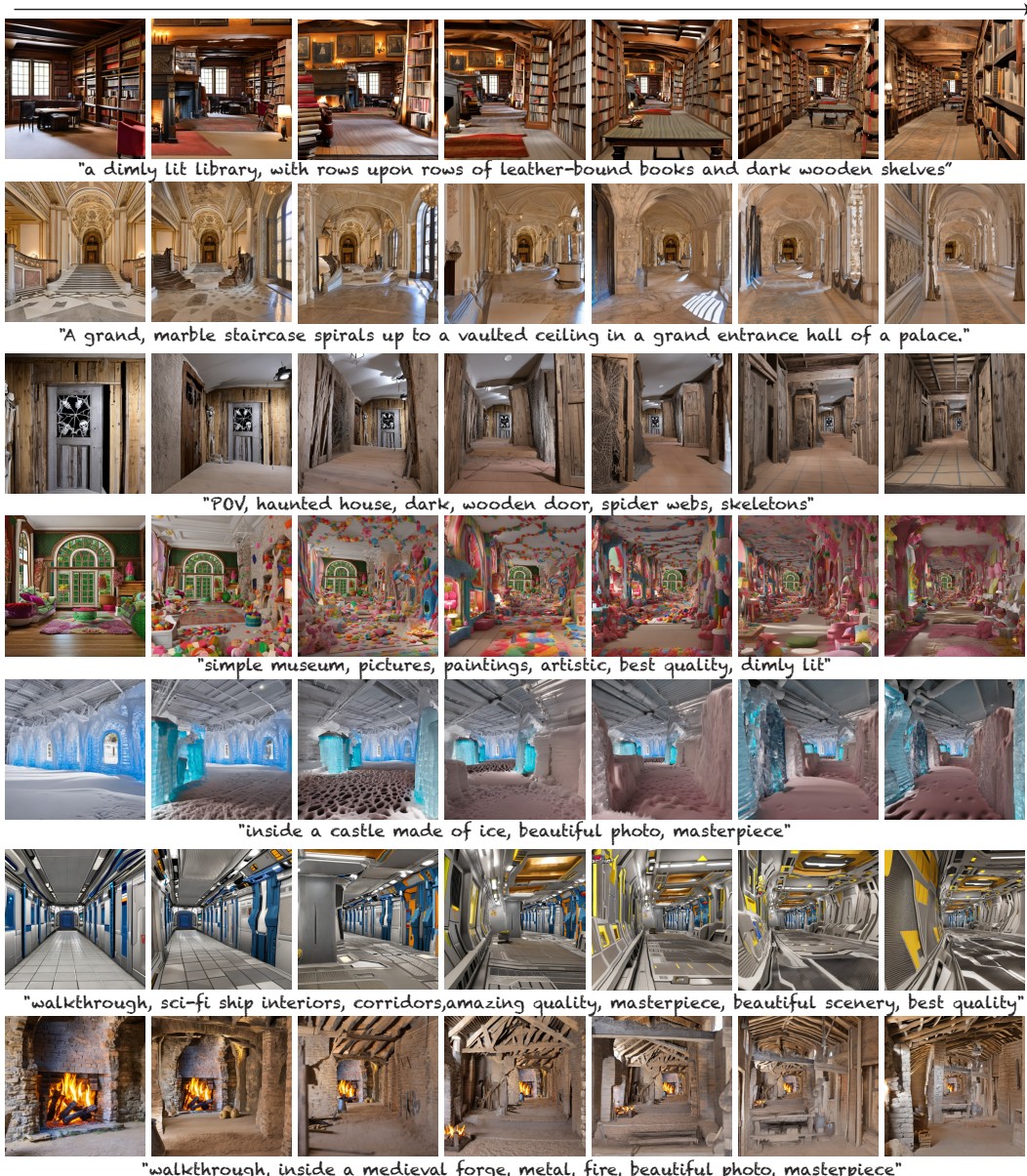

"a dimly lit library, with rows upon rows of leather-bound books and dark wooden shelves"

"A grand, marble staircase spirals up to a vaulted ceiling in a grand entrance hall of a palace."

"POV, haunted house, dark, wooden door, spider webs, skeletons"

"simple museum, pictures, paintings, artistic, best quality, dimly lit"

"inside a castle made of ice, beautiful photo, masterpiece"

"walkthrough, sci-fi ship interiors, corridors,amazing quality, masterpiece, beautiful scenery, best quality"

"walkthrough, inside a medieval forge, metal, fire, beautiful photo, masterpiece"

Figure 4: Sample frames from our generated videos. Our method generates high-quality and diverse scenes while following a desired camera trajectory. All videos are included in SM.

as imaginary scenes. We generated 50-frame long videos, with a fast-moving camera motion, which follows a backward motion combined with rotations of the camera (see Sec. A of appendix for more details). Since our method represents the scene via a mesh, we focus mainly on indoor scenes. Synthesizing outdoor scenes would require careful handling in cases of dramatic depth discontinuities (as discussed in Sec. 5).

Figure 1 and Fig. 4 show sample frames from our videos. The results demonstrate the effectiveness of our method in producing high-quality and geometrically-plausible scenes, depicting significant parallax, complex structures, and various complex scene properties such as lighting or diverse materials (e.g., ice). Additional results, videos, and output depth maps are included in the Supplementary materials (SM).

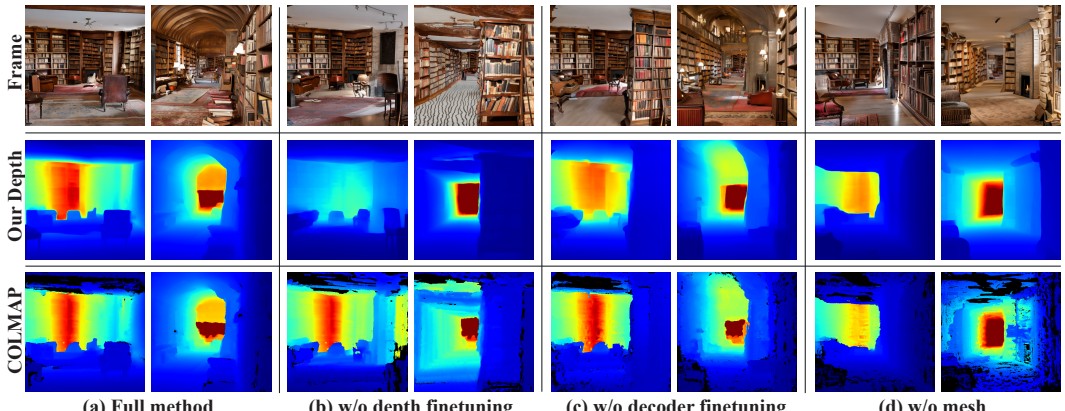

**(a) Full method**  **(b) w/o depth finetuning**  **(c) w/o decoder finetuning**  **(d) w/o mesh**

Figure 5: **Sample ablation results** of (a) our full method, (b) w/o depth finetuning (Sec. 3.3), (c) w/o decoder finetuning (Sec. 3.4) and (d) w/o mesh (Sec. 3.1). We present sample frames, output depth maps, and COLMAP's depth maps. As seen, all components are essential for obtaining high-quality and 3D-consistent videos.

| | Rot. (deg.) ↓ | Trans. (%) ↓ | Reproj. (pix)↓ | SI-RMSE ↓ | Density (%) ↑ | CLIP-AS ↑ | AMT(%)↑ |
|---|---|---|---|---|---|---|---|
| Ours | **0.61** | **0.01** | **0.33** | **0.17** | **0.91** | **5.85** | - |
| w/o depth f. | 2.24 | 0.02 | 0.37 | 0.41 | 0.90 | 5.75 | 52.7 |
| w/o decoder f. | 2.81 | 0.02 | 0.82 | 0.17 | 0.87 | 5.74 | 63.0 |
| w/o mesh | 3.55 | 0.04 | 0.57 | 0.18 | 0.78 | 5.73 | 69.1 |
| Warp-inpaint | 4.89 | 0.09 | 0.97 | 0.67 | 0.70 | 5.43 | 85.6 |

Table 1: **Ablations.** For each ablation baseline, we report, from left to right: accuracy of estimated camera poses, reprojection error, depth SI-RMSE, and the percentage of reconstructed video pixels. We also report CLIP aesthetics score [1] and measure visual quality using AMT user study, where we report the percentage of judgments in favor of our full method. See Sec. 4.1 and Sec. 4.2 for more details.

## 4.1 Metrics

In all experiments, we use the following metrics for evaluation:

**Depth consistency.** We evaluate the depth consistency of a video by applying COLMAP, using known camera poses, and measuring the Scale-Invariant Root Mean Square Error (SI-RMSE) [14]:

$$\text{SI-RMSE} = \sqrt{\frac{1}{N^2} \sum_{\mathbf{p},\mathbf{q}\in I} [(\log D^o(\mathbf{p}) - \log D^o(\mathbf{q})) - (\log D^c(\mathbf{p}) - \log D^c(\mathbf{q}))]^2} \quad (7)$$

where $D^o$ and $D^c$ are our output depth maps and COLMAP depths, respectively. We compute the SI-RMSE for each frame and report the average per video. To assess the completeness of the reconstruction, we also measure the percentage of pixels recovered by COLMAP, i.e., passed its geometric consistency test. In addition, we report the average reprojection error of COLMAP reconstruction, which indicates the consistency of the video with COLMAP's point cloud.

**Pose accuracy.** We measure the accuracy of the estimated camera trajectory. Specifically, we provide COLMAP only with intrinsic parameters and align the two camera systems by computing a global rotation matrix [61]. We measure the normalized mean accuracy of camera positions (in percentage, relative to the overall camera trajectory), and camera rotations (in degrees).

**Visual quality.** We use Amazon Mechanical Turk (AMT) platform for evaluating the visual quality of our videos compared to the competitors. We adopt the Two-alternative Forced Choice (2AFC) protocol, where participants are presented with two side-by-side videos, ours and the competitor's (at random left-right ordering) and are asked: "*Choose the video/image that has better visual quality, e.g., sharper, less artifacts such as holes, stretches, or distorted geometry*". Additionally, we compute the CLIP aesthetic score [1]–an image aesthetic linear predictor learned on top of CLIP embeddings.

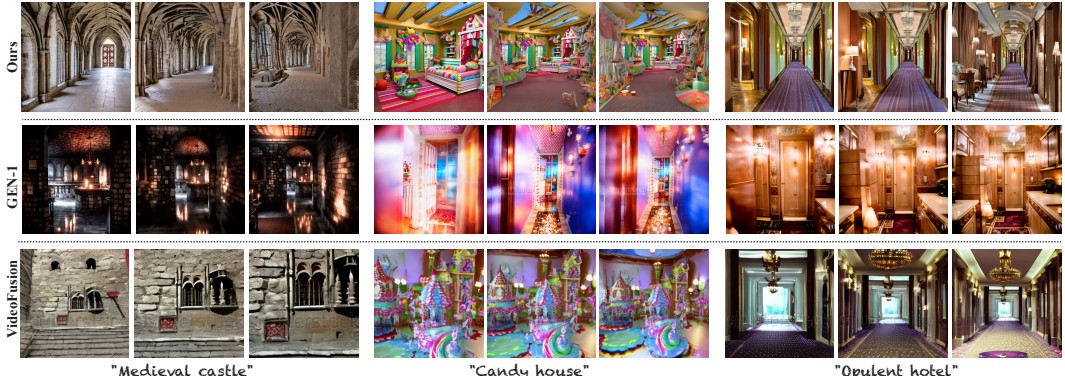

Figure 6: **Baseline comparison.** Sample results from our method (top), GEN-1 [15] (middle) and VideoFusion [32] (bottom) on 3 different prompts. Other methods exhibit saturation and blurriness artifacts in addition to temporally inconsistent frames.

## 4.2 Ablation

We ablate the key components in our framework: (i) depth finetuning (Sec. 3.3), (ii) decoder finetuning (Sec. 3.4), and (iii) our unified mesh representation (Sec. 3.1). For the first two ablations, we apply our method without each component, and for the third, we replace our mesh with 2D rendering, i.e., warping the content from the current frame to the next, using a depth-aware splatting [37]. In addition, we consider the baseline of simple warp-inpaint, i.e., using splatting with the original estimated depth (w/o finetuning). We ran these ablations on 20 videos, using a set of 10 diverse text prompts. See Sec. B for the list of prompts and SM sample videos.

Table 1 reports the above COLMAP metrics (see Sec. 4.1). Our full method results in notably more accurate camera poses, denser reconstructions, and better-aligned depth maps. Excluding depth finetuning significantly affects the depth consistency (high SI-RMSE). Excluding decoder finetuning leads to high-frequency temporal inconsistency, which leads to higher reprojection error. Without a mesh, the reconstruction is significantly sparser and we observe a drop in all metrics. These results are illustrated in Fig. 5 for a representative video.

To assess visual quality, we follow the AMT protocol described in Sec. 4.1, where we ran each test w.r.t. one of the baselines. We collected 4,000 user judgments from 200 participants over 20 videos. As seen in Table 1, the unified mesh plays a major role in producing high-quality videos, where the judgments were ambiguous w.r.t w/o depth finetuning baseline. We hypothesize this is because this baseline still leverages the mesh representation, thus the visual artifacts are temporally consistent, making them more difficult to perceive.

## 4.3 Comparison to Baselines

To the best of our knowledge, there is no existing method designed for our task – text-driven perpetual view generation. We thus consider the most relevant methods to our setting:

- **VideoFusion (VF) [32].** A text-to-video diffusion model[2], which takes as input a text prompt and produces a 16-frame long video at $256 \times 256$ resolution.

- **GEN-1 [15].** A text-driven video-to-video translation model, where the translation is conditioned on depth maps estimated from an input source video. Note that in our setting the scene geometry is unknown and is constructed with the video, whereas GEN-1 takes the scene geometry as input, thus tackling a simpler task.

Due to the inherent differences between the above methods, we follow a different evaluation protocol in comparison with each one of them.

---

[2]To the best of our knowledge, this is the only publicly available text-to-video model.

|  | Rot. (deg.) ↓ | Trans. (%) ↓ | Reproj. (px) ↓ | Density (%) ↑ | CLIP-TS ↑ | AMT (%) ↑ |
|---|---|---|---|---|---|---|
| Ours | - | - | **0.29** | **81.4** | **0.25** | **95.97** |
| VF | - | - | 0.78 | 46.36 | 0.22 | |
| Ours | **1.71** | **3.94** | **0.60** | **91.72** | **0.26** | **69.34** |
| GEN-1 | 2.47 | 8.09 | 1.64 | 53.96 | **0.26** | |

Table 2: **Comparison to baselines.** (top) VideoFusion [32], and (bottom) GEN-1 [15]. We evaluate 3D consistency using COLMAP metrics and measure visual quality using AMT user study, where we report the percentage of judgments in our favor (see Sec. 4.2). Our method outperforms the baselines on all metrics. Note that the experiential setup for these two baselines is different (see Sec. 4.3).

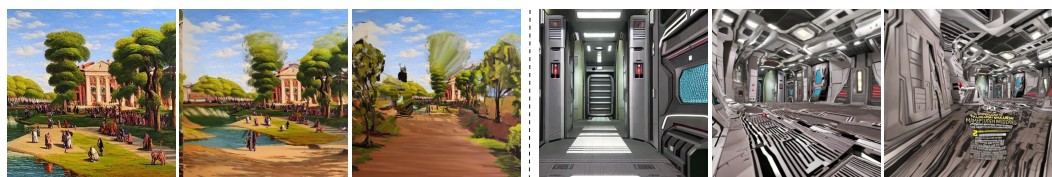

Figure 7: **Limitation.** Outdoor scenes are poorly represented with our mesh representation (left). Occasionally we may observe quality degradation due to error accumulation (right).

### 4.3.1 Evaluating 3D-consistency

To compare to VF, we used the same set of prompts as in Sec. 4.2 as input to their model and generated 1000 videos. We evaluate the 3D consistency of VF's videos and ours, using the same metrics as in Sec. 4.2, where we used COLMAP without any camera information, and downsampled our videos to match their resolution. As seen in Table 2, our method significantly outperforms VF in all metrics. This demonstrates the effectiveness of our mesh representation in producing 3D-consistent videos, in contrast to VF, which may learn geometric priors by training on a large-scale video dataset. Note that since VF does not follow a specific camera path, measuring its camera pose accuracy is not possible.

GEN-1 is conditioned on a source video, while our method takes camera trajectories as input. Thus, we used the RealEstate10K dataset [66], consisting of curated Internet videos and corresponding camera poses.

We automatically filtered 22 indoor videos that follow a smooth temporal backward camera motion to adapt it to our method's setting. The camera poses are used as input to our method, whereas the RGB frames are used as input video to GEN-1 while supplying both methods with the same prompt. We generate 110 videos[3] using five different prompts from our ten prompts set that allow reasonable editing of a house (such as "library" rather than "cave"). See Sec. B of appendix for more details.

We measure the video's 3D consistency metrics, as discussed in Sec. 4.2, and report them in Table 2. Our method outperforms GEN-1 in all metrics, even though the geometric structure of the video (depth maps) was given to GEN-1 as input.

### 4.3.2 Visual Quality and Scene Semantics

We conducted an extensive human perceptual evaluation as described in Sec. 4.1. We collect 3,000 user judgments over 60 video pairs from 50 participants. The results of this survey w.r.t. each method are reported in Table 2. As seen, when compared to VF, our method was preferred in 96% of the cases and in 69% of the cases in comparison with GEN-1. Sample results of those comparisons can be seen in Fig. 6. Note that since GEN-1, VF, and StableDiffusion do not exhibit the same underlying data distribution, existing visual quality metrics such as FID [17] cannot be used.

To measure how well our generated videos adhere to the desired scene, we measure the CLIP text-similarity score - mean cosine similarity between the prompt and each of the frames. As seen in Table 2, our method follows the scene text prompt more closely than VF and is on par with GEN-1.

---

[3]There is only a limited UI access to GEN-1, which restricts the scale of this experiment compared to VF.

# 5  Discussion and Conclusion

We posed the task of text-driven 3D-consistent perpetual view generation and proposed a test-time optimization approach to generate long-term videos of diverse scenes. Our method demonstrates two key principles: (i) how to harness two powerful pre-trained models, allowing us to generate scenes in a zero-shot manner without requiring any training data from a specific domain, and (ii) how to combine these models with a unified 3D scene representation, which by construction ensures feasible geometry, and enables high quality and efficient rendering. As for limitations, the quality of our results depends on the generative and geometric priors and may sometime decrease over time due to error accumulation (Fig. 7 right). In addition, since we represent the scene with triangular meth, it is difficult for our method to represent dramatic depth discontinuities, e.g., sky vs. ground in outdoor scenes (Fig. 7 left). Moreover, since our method works in an online fashion and does not change the previously generated frames, the errors that appeared in previous frames can propagate to the next frames. Devising a new 3D scene representation tailored for the perpetual view generation of arbitrary scenes is an intriguing avenue of future research. We believe that the principles demonstrated in our work hold great promise in developing lightweight, 3D-aware video generation methods.

# 6  Acknowledgments

We thank Shai Bagon for his insightful comments. We thank Narek Tumanyan for his help with the website creation. We thank GEN-1 authors for their help in conducting comparisons. This project received funding from the Israeli Science Foundation (grant 2303/20).

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

# A Implementation Details

We provide implementation details for our framework and finetuning/generation regime.

**Runtime.** We use the Stable Diffusion [48] model that was additionally finetuned on the inpainted task to perform inpainting. We use DDIM scheduler [53] with 50 sampling steps for each generated frame. Synthesizing 50 frame-long videos with our full method takes approximately 2.5 hours on an NVIDIA TeslaV100 GPU. Specifically, Table 3 reports runtime required for each frame.

| Rendering | Inpainting | Depth model finetuning | Decoder finetuning |
|-----------|------------|------------------------|--------------------|
| ~40 sec | ~5 sec | ~40 sec | ~60 sec |

Table 3: **Runtime per-frame.** Number of seconds required for each step of the framework. Note that Rendering includes antialiasing and floating artifacts fix steps. In comparison, VideoFusion [32] takes 15 seconds for a 16-frame video of 256*256 resolution on an NVIDIA TeslaV100 GPU. Since we do not have direct access to the GEN-1 [15] model, we are limited in assessing its runtime. To get a rough estimate, we measured their online interface runtime, which takes 1 minute to edit a video with 25 frames.

**Depth prediction model and LDM decoder finetuning.** We use MiDaS-DPT Large [42] as our depth prediction model. For each generated frame, we finetune it for 300 epochs, using Adam optimizer [25] with a learning rate of $1e-7$. Additionally, we revert the weights of the depth prediction model to the initial state, as discussed in Sec. 3.3. We finetune the LDM decoder for 100 epochs on each generation step using Adam optimizer with a learning rate of $1e-4$.

**Camera path.** Our camera follows a rotational motion combined with translation in the negative depth direction. Starting from a simple translation for $k$ frames, every $n$ frames we randomly sample a new rotation direction in the $x$-$z$ plane (panning), that camera follows for $n$ frames. In our experiments, we set $k = 5$ and $n = 5$. We use PyTorch3D [44] to render and update our unified 3D representation.

**Mask handling.** We observed that the scene's geometry sometimes induces out-of-distribution inpainting masks for the Stable Diffusion inpainting model. To address this issue, we perform a morphological open operation on the inpainting mask: $\boldsymbol{M_O} = Open(M)$ with kernel size 3. Then we inpaint the mask difference $\boldsymbol{M} - \boldsymbol{M_O}$ using Telea [55], while the inpainting model operates on $\boldsymbol{M_O}$ afterward.

## A.1 Mesh update.

As discussed in Sec. 3.5 in the paper, we update the mesh as follows: given an image $\boldsymbol{I_{t+1}}$, a mask of pixels to unproject $\boldsymbol{M}$, a corresponding depth map $\boldsymbol{D_{t+1}}$ and a camera pose $\boldsymbol{C_{t+1}}$, we unproject the content in a process that is denoted in the main paper by $\tilde{\mathcal{M}}_{t+1} = UnProj\left(\boldsymbol{M}, \boldsymbol{I_{t+1}}, \boldsymbol{D_{t+1}}, \boldsymbol{C_{t+1}}\right)$. First, each pixel center is unprojected by its depth value and camera pose into a 3D mesh vertex with the pixel's color as its vertex color. Then, each unprojected four neighboring pixels with coordinates $(i, j), (i+1, j), (i, j+1), (i+1, j+1)$ are used to define two adjacent triangle faces.

To prevent holes in the generated mesh we connect the existing mesh $\mathcal{M}_t$ to the newly unprojected part $\tilde{\mathcal{M}}_{t+1}$ by inserting additional triangles. To do this, we first extract the boundary pixels surrounding the inpainting mask $\boldsymbol{M}$, and find the faces from $\mathcal{M}_t$ that are projected to the current view $\boldsymbol{C_{t+1}}$. For each such face, we select the 3D point closest to the current camera center: $p^* = \underset{p \in (p_1, p_2, p_3)}{\arg\min} \|p - c\|^2$, where $c$ is the camera center of $\boldsymbol{C_{t+1}}$, and $(p_1, p_2, p_3)$ are the points of a given triangle. Finally, we add the selected points to the triangulation scheme described above, which automatically creates the required triangles connecting $\mathcal{M}_t$ and $\tilde{\mathcal{M}}_{t+1}$.

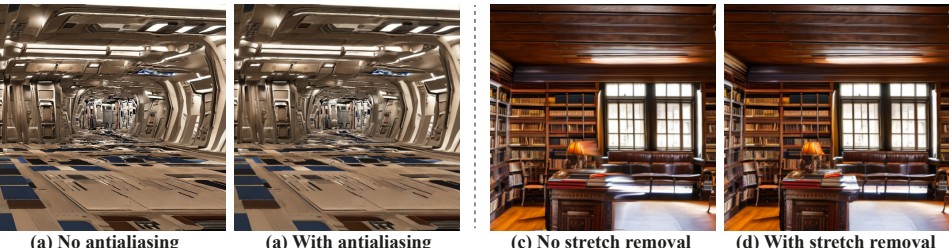

| (a) No antialiasing | (a) With antialiasing | (c) No stretch removal | (d) With stretch removal |

Figure 8: **Rendering improvements.** Left: effect of rendering without antialiasing (a) and with antialiasing (b). Right: effect of rendering without stretched triangles removal (a) and with stretched triangles removal (b).

## A.2 Rendering

**Antialiasing.** Rendering an image from a mesh requires projecting points and faces to a 2D plane. The rasterization process often creates aliasing artifacts (Fig. 8, left), especially when a high-resolution mesh content from an earlier frame is rendered to a later frame, resulting in a large amount of triangle that needs to be rasterized into a low number of pixels. To avoid these artifacts, we apply antialiasing, similar to the image resize antialiasing method - we render the mesh at x2 higher resolution, apply Gaussian blur, and resize it to the required resolution.

**Stretched triangles removal.** As described in Sec. 3.6, the stretched triangles are forming between close and far away content along regions of depth discontinuities (Fig. 8, right), and we would like to remove them. Following Liu et al. [29], we apply the Sobel filter on the depth map $D_t$ to detect regions of depth discontinuities and threshold the values below the threshold of 0.3. Then we find triangles that are projected to the selected edge regions and filter them based on their normals. Specifically, we keep only the following triangles: $\{tr_i|(\text{center}(tr_i) - c)^T n < \epsilon\}$, where $\epsilon$ is the threshold, $n$ is the normal of a triangle, and $c$ is the camera center of $C_{t+1}$. In practice, we set $\epsilon = -0.05$.

**Floating artifacts fix.** As described in Sec 3.6, content at the border of the current frame can be mapped towards the interior of the next frame due to parallax. This creates "floating artifacts," shown in Fig. 3 in the paper. To overcome it, we pad the previous depth map $D_t$ with border depth values to 1.5x the rendering resolution, as if content exists beyond the image borders. Then, after we warp the padded depth to the next camera location, we get a new mask, $M_{pad}$, that contains the content beyond the image borders. We use this mask to mask out the floating regions and thus enable the inpainting model to fill those holes with closer content. We perform this procedure on the image that was already rendered in 2x resolution due to the antialiasing step described above.

## B Baseline Comparison Details

**VideoFusion.** Since VideoFusion [32] does not have an explicit way to control the motion presented in a video, we append "zoom out video" and "camera moving backward" to the input prompt to encourage the generated videos to follow a backward camera motion. We generate 1000 videos of 16 frames each, in 256 resolution. When comparing our method to theirs, we downsample our videos to 256 resolution. In this comparison, we use the full set of 10 prompts:

1. "*indoor scene, interior, candy house, fantasy, beautiful, masterpiece, best quality*"
2. "*POV, haunted house, dark, wooden door, spider webs, skeletons*"
3. "*walkthrough, an opulent hotel with long, carpeted hallways, beautiful photo, masterpiece, indoor scene*"
4. "*A dimly lit library, with rows upon rows of leather-bound books and dark wooden shelves*"
5. "*walkthrough, inside a medieval castle, metal, beautiful photo, masterpiece, indoor scene*"
6. "*Simple museum, pictures, paintings, artistic, best quality, dimly lit*"
7. "*walkthrough, sci-fi ship interiors, corridors,amazing quality, masterpiece, beautiful scenery, best quality*"

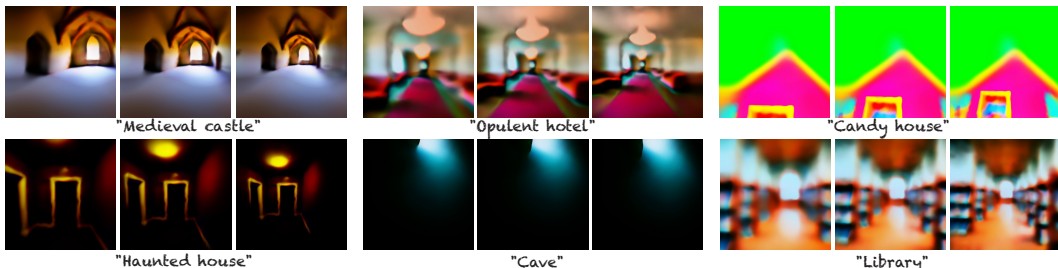

Figure 9: **StableDreamFusion sample results.** We provide it our camera trajectory and a text prompt "*a DSLR photo of the inside a \**". Full results can be found in the SM.

8. "*A grand, marble staircase spirals up to a vaulted ceiling in a grand entrance hall of a palace.A warm glow on the intricately designed floor*"
9. "*POV, cave, pools, water, dark cavern, inside a cave, beautiful scenery, best quality*"
10. "*inside a castle made of ice, beautiful photo, masterpiece*"

**GEN-1**  To compare to GEN-1 [15], we used the RealEstate10K dataset [66], consisting of curated Internet videos and corresponding camera poses. We filtered 22 indoor videos that follow a smooth temporal backward camera motion to adapt it to our method's setting. To do that, we filter videos that adhere to the following constraints:

$$\frac{(c_{t+1} - c_t)^T v_t}{\|(c_{t+1} - c_t)\| \cdot \|v_t\|} \geq 0.9 \quad \forall t = 1, \dots, \text{n\_frames},$$

where $c_t$ is the center of the camera at frame $t$, and $v_t$ is the viewing direction (last column of the rotation matrix). We subsample the filtered videos to contain 25 frames and reverse the video if it had a positive average displacement $(c_{t+1} - c_t)$ in the $z$ direction.

Since MiDaS produces depth maps that have a different range compared to the depth assumed in the RealEstate10K videos, we can't directly take the camera extrinsics from the RealEstate10K [66] data. To align those ranges, we need to find a scaling factor to multiply the MiDaS predictions with. To do that we run COLMAP and compute depth maps, using its dense reconstruction. Our scaling factor is then the ratio of median depth values of COLMAP and MiDaS:

$$r = \frac{\text{median}(\{D_t^C\}_{t=1}^N)}{\text{median}(\{D_t^M\}_{t=1}^N)}, \tag{8}$$

where $D_t^C$ and $D_t^M$ are the COLMAP and MiDaS depth maps for frame $t$ respectively, and $N$ is the number of frames in the video.

For comparison to GEN-1 we used prompts #1-#5 from the above prompt list that allow reasonable editing of a video, depicting an interior of a house.

**StableDreamFusion.**  In addition, we compare our method to StableDreamfusion [54], an open-source text-to-3D model capable of generating an implicit function of a scene (NeRF [35]) from text prompts. We provide it our camera trajectory and a simple text prompt such as "*a DSLR photo of the inside a cave*" (when providing prompts from our usual prompts set, StableDreamFusion was unable to converge to meaningful results). The generated scenes contain a lot of blur and unrealistic artifacts, as can be seen in Figure 9 since to achieve good visual quality, NERF requires multiple viewpoints of a scene from different angles. Full video results of StableDreamFusion can be found in the SM.

## C  Broader impact

Our framework is a test-time optimization method, which does not require any training data. Nevertheless, our framework uses two pre-trained models: a depth prediction model and a text-to-image diffusion model [48], which are susceptible to biases inherited to their training data (as discussed in Sec. 5 in [48]). In our context, these models are used to generate 3D static scenes. To avoid harmful content, we consider only text prompts that describe general scenery and objects, while avoiding prompts that involve sensitive content such as humans.

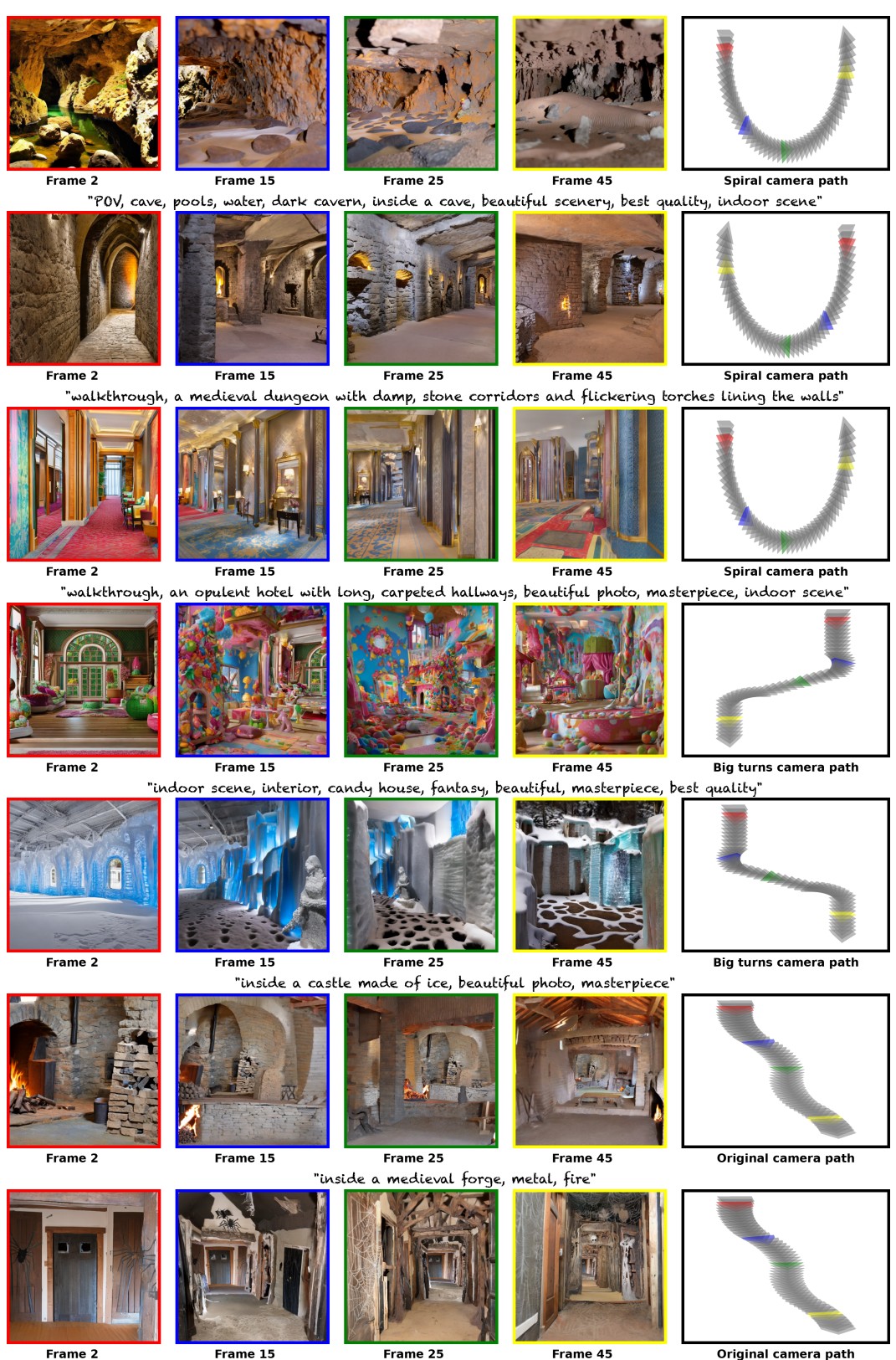

Figure 10: **Additional results with different camera motion.** Left: Representative frames generated by our method with colorful borders corresponding to specific camera locations. Right: The camera path used in the generated video, with cameras color-coded to match the displayed frames' borders, reflecting their respective locations along the motion path.

