# OpenReview forum: "SceneScape: Text-Driven Consistent Scene Generation"
_NeurIPS.cc/2023/Conference — NeurIPS 2023 poster_

### Official Review · Reviewer_GkJU · 2023-06-23

**Soundness:** 2 fair
**Presentation:** 3 good
**Contribution:** 2 fair
**Rating:** 4
**Confidence:** 4

**Summary:**

The paper present a method for text-driven perpetual view generation. It takes as input a text-prompt and carema trajectory as input, optimizing a unified mesh representation for rendering coherent video sequence accordingly. The key idea is straight-forward by connecting text-driven inpainting and depth warping with mesh-based representation to ensure 3D consistency.

**Strengths:**

- The paper proposes the first method for text-driven perpetual view generation with promising results, i.e., diverse scene textures and inter-frame coherence.
- As an emerging direction with limited open-source work for comparisons, the authors have tried their best to compare the proposed method with others from different perspective, i.e., video generation, text-drive vid2vid, text-to-3d, for extensive evaluations.
- The paper is well-organized and easy to follow.

**Weaknesses:**

- Only bounded indoor scene as the model is constrained by accuracy of depth map. However, outdoor unbounded scene is difficult to generate since depths has infinite ranges and discontinuities.
- Moreover, the proposed method has the strong tendency to generate tunnel-specific scenes, i.e., a narrow hallway or tunnel through the space. It might be caused by the inpainting+depth continuity assumption. Because the surrounding walls acts as strong geometry cues for view consistency as camera moving backward.
- Prompt engineering is required. Keywords like “POV” and “walkthrough” is critical for the generation results. The author should discuss their effects and design choices carefully as the paper claims using “free-vocabulary text”.
- I may question the second contribution in the introduction, which claims the proposed method as the first zero-shot scene generation method. Unlike zero-shot text-to-3D object generation (DreamField, Dreamfusion, etc.) where a 3D representation is generated and camera can freely move, the proposed method binds the camera movement with the optimization process, which limits the camera trajectory after the optimization done. Therefore, it is only a 3D-aware video generation instead of 3D scene.
- This is a personal one. The general idea (text-to-image + depth inpainting) of this method is not completely novel, with lots of engineering tricks to make it work. And I believe some details are missed to evaluate other technical parts like depth & decoder finetuning. Please check the Question section.
- I believe the work miss some important literature for reference, in terms of zero-shot text-driven synthesis and indoor scene synthesis.
    - zero-shot text-driven synthesis:

        For optimization-based one, there are StyleCLIP and etc. For training without text-image pair, there are LAFITE.

        StyleCLIP: Text-Driven Manipulation of StyleGAN Imagery, Patashnik et al, ICCV 2021.

        LAFITE: Towards Language-Free Training for Text-to-Image Generation, Zhou et al, CVPR 2022.

    - scene synthesis / language-driven scene synthesis:

        Learning spatial knowledge for text to 3D scene generation, Chang et al, EMNLP 2014

        Language-driven synthesis of 3d scenes from scene databases, Ma et al, SIGGRAPH Asia 2018.

        Text2Light: Zero-shot Text-driven HDR Panorama Generation, Chen et al, SIGGRAPH Asia 2022.
- typos:
    - lacks a full stop in #L126
    - Figure or Fig. is not consistent in the paper. For example, #L195.
    - ‘mesh’ instead of ‘meth’ in #L285

**Questions:**

- Why finetune depth prediction model at test time? Finetuning may overfit/forgetting when the generated video is too long. Plus, the error would accumulate along with time steps. When I come to the ablation study of depth finetuning, I found that the RGB texture dose not jittering across video frame. In that case, does the depth map finetuning matter?
- I’m not clear how depth finetuning really work as g is reverted at each time step.
- The initialization of mesh M_0 is not clear to me. Does the number of point cloud is consistent with the nubmer of pixels? Moreover, how to get faces and edges of mesh from point cloud?
- Online mesh update: the merging in Eq. 6 would lead to conflict. How to solve conflict is not introduced.
- Figure 7 shows the failure cases resulted by error accumulation. Here comes the question, can authors evaluate the robustness and error rate of the method? For example, given a fixed set of text prompts, how many scenes can be successfully generated?
- Moreover, I would like to see experiments or discussion of sensitivity to hyperparameters.

**Limitations:**

Limitations and negative societal impact are discussed.

---

> ### Author Rebuttal · Authors · 2023-08-09
>
> We appreciate the comments and feedback from the reviewer.
>
> ### Outdoor scenes
> Extending our method to handle outdoor scenes is an interesting research direction that indeed poses several challenges, including the accuracy of the estimated depth, as pointed out by the reviewer, as well as handling dramatic depth discontinuities. We discuss this in Section 5 of the paper.
>
> ### Camera Motion
> Please refer to our global response, where we elaborate on our camera motion configuration and include additional results.
>
> ### Prompt selection
> Our method does not require careful prompt engineering and can work with input prompts that describe general, static indoor scenes. We appended a few words such as “masterpiece, best quality” to improve the visual quality and the fidelity to the text prompt, which is common practice in numerous papers [61,62,63]. We omitted these words and re-run our method. As seen in the attached PDF and in the third row of the provided video, our method is agnostic to these specific words and works well with very simple scene descriptions. We will clarify this in the paper.
>
> [61] Ben Poole, Ajay Jain, Jonathan T. Barron, Ben Mildenhall: DreamFusion: Text-to-3D using 2D Diffusion. ICLR 2023
>
> [62] Alexander Quinn Nichol, Prafulla Dhariwal, Aditya Ramesh, Pranav Shyam, Pamela Mishkin, Bob McGrew, Ilya Sutskever, Mark Chen: GLIDE: Towards Photorealistic Image Generation and Editing with Text-Guided Diffusion Models. ICML 2022
>
> [63] Aditya Ramesh, Prafulla Dhariwal, Alex Nichol, Casey Chu, Mark Chen: Hierarchical Text-Conditional Image Generation with CLIP Latents. CoRR abs/2204.06125
>
>
> ### 3D-Consistent video generation
> Zero-shot text-to-3D object generation methods such as DreamFusion are not suitable for the task of perpetual view generation. This is demonstrated in Figure 9 in the SM, where we provided DreamFusion with our camera poses and input text prompt. As seen, this method completely fails to optimize a high-quality NeRF. Generally, such methods require multiple observations of the same scene point to coverage, which are not available in our setting.
>
> ### Literature reference
> We thank the reviewer for the references. Although these works are not directly aimed at the task of video generation, we will include them in the context of zero-shot text-driven content manipulation/generation.
>
> ### Necessity of depth model finetuning
> As described in Section 3.3 in the paper and demonstrated in our ablation study, per-frame monocular depth predictions tend to be temporally inconsistent, even across nearby video frames. Finetuning the depth model to align the depth predictions across frames (Eq. 3) mitigates this issue. Note that the finetuned depth prediction model can still retain its prior since (i) the finetuning is done only for 300 steps, and (ii) we revert the model weights to its original non-tuned ones for each frame. Thus, we do not deviate significantly from its prior, thus avoiding catastrophic forgetting. We refer the reviewer to previous works [24,64,65] in the context of video depth estimation for a more elaborated discussion about test-time depth finetuning.
>
> [64] Xuan Luo, Jia-Bin Huang, Richard Szeliski, Kevin Matzen, Johannes Kopf: Consistent video depth estimation. ACM Transactions in Graphics. 39(4)
>
> [65] Zhoutong Zhang, Forrester Cole, Richard Tucker, William T. Freeman, Tali Dekel: Consistent depth of moving objects in video. ACM Transactions in Graphics 40(4)
>
> ### Mesh initialization and update
> As described in Section 3.1 of the paper, we initialize the mesh by unprojecting each pixel in the first frame into a corresponding 3D point. Then, each unprojected four neighboring pixels are used to define two adjacent triangle faces. We refer the reviewer to Section 1.1 in the SM for details about the mesh update. Note that at each update step (Section 3.5 in the paper), we only add the newly generated content while keeping the existing content in the mesh, thus no conflict is introduced.
>
> ### Error accumulation
>
> Following the reviewer’s comment, we measured our error accumulation for our video evaluation set. To do so, we measured: (i) visual quality by measuring the CLIP aesthetic score as a function of frame time step and (ii) geometric consistency by evaluating COLMAP on the first $n$ frames ($n$=10,20,30,40,50) and computed the corresponding SI-RMSE as a representative metric for the geometric consistency.
>
> As can be seen in the attached PDF, our method exhibits slight degeneration in visual quality, as expected and discussed in Section 5 in the paper, yet results in a significant improvement over the warp-inpaint baseline. In terms of geometry consistency, our SI-RMSE remains steady or even slightly improves with the length of the video (since COLMAP works better when more correspondences can be established).
>
>
> ### Hyperparameters sensitivity
>
> Since we did not perform any extensive hyperparameters search and used the same hyperparameters to generate all of our results, we omitted any analysis of the hyperparameters' sensitivity.

---

> > ### Comment · Reviewer_GkJU · 2023-08-15
> >
> > Thanks for the detailed responses. The rebuttal from authors address my concerns regarding camera motions and prompts engineering. However, I do find that the "big turns" camera motions lead to repeated patterns and degradation of generated results. Thus, it seems like the method only solve the problem of text-driven perpetual view generation. I would like the claims like 3D scene generation to be removed in the revised version in case the the paper is accepted.
> >
> > A follow-up question on mesh initialization. As the author responded, there is no update to the existing mesh. The error accumulated by depth estimator will always exist in the mesh representation. Look forward to the comment on this issue.

---

> > > ### Author Response · Authors · 2023-08-16
> > >
> > > Thank you for the feedback. We agree that our method is aimed at perpetual view generation and we will revise the exposition as suggested. Specifically, replacing “3D-consistent scene generation” with “3D-consistent perpetual view generation”.
> > >
> > > Regarding the sharp turns results, following the reviewer’s comment, we measured the visual quality and the geometric consistency for the videos presented in the attached video, using the same metrics described in Section 4.1 of the paper:
> > >
> > > | Rot. (deg.) ↓ | Trans. (%) ↓ | Reproj. (pix) ↓ | SI-RMSE ↓ | Density (%) ↑ | CLIP-AS ↑|
> > > |---------------|--------------|-----------------|-----------|---------------|---------------|
> > > | 0.1079        | 0.0012       | 0.2135          | 0.1407    | 0.9225        | 5.8379|
> > >
> > > These results suggest that our videos with bigger rotations (“Big turns”) are comparable to our default setting.
> > > Regarding the mesh update, indeed our method works in an online fashion and does not change the previously generated frames. Thus, as the reviewer pointed out, the errors that appeared in previous frames, can propagate to the next frames. We will add this discussion to our limitation section and include the error accumulation evaluation in the revised version of the paper.

---

> > > > ### Comment · Reviewer_GkJU · 2023-08-19
> > > >
> > > > Thanks for your detailed response and additional experiments. Both the pros and cons of the proposed method are clearly discussed. I do not have other concerns so far.

---

### Official Review · Reviewer_Zj8B · 2023-06-26

**Soundness:** 2 fair
**Presentation:** 3 good
**Contribution:** 2 fair
**Rating:** 3
**Confidence:** 4

**Summary:**

This paper approaches the task of text-conditioned generation of a walkthrough video upon (mostly indoor) scenes. It proposes to leverage pretrained text-to-image and pretrained depth prediction models, to generate these videos without finetuning. Implementation generates videos in an online manner by repeated warping, inpainting and mesh aggregation. Contributions include test-time overfitting of depth and inpainting, and experiments show these are important to geometric  and semantic performance.

**Strengths:**

This paper uses a cool insight to generate 3D walkthroughs using 2D pretrained models

-	The paper leverages the power of large-pretrained models for online generation of many images in a consistent manner, without the need for finetuning. Careful implementation enables these images to create a consistent scene

-	Videos are long (50 frames with high movement), showing impressive consistency. They are also typically visually appealing (especially compared to a naïve implementation)

Several contributions are executed to make performance high-quality

-	Finetuning depth and decoder modules with previously generated content meaningfully improves consistency

-	Dealing with floating artifacts is a clear challenge that the proposed method deals with effectively (Fig 3)

-	The method performs a union operation on mesh. This is not novel alone, but (correct) implementation is important to performance.

-	In total, the proposed method improves COLMAP fitting errors by 3-6x over naïve method

**Weaknesses:**

The method appears to be limited to backward movement, with little rotation. This trajectory setup makes the task easier and less interesting, as it does not handle side-to-side movement, or “looking around” which are important components of walkthroughs e.g. see RealEstate10K

-	Consider the motivation for this paper in L15-16 “By observing only a single photo of a scene, we can imagine what it would be like to explore it, move around, or turn our head and look around”. Unfortunately, this work does not answer this problem satisfactorily. Instead, it produces nice-looking and consistent results as we move backwards through an infinitely-long corridor, where we only slightly turn our head periodically.

-	More “looking around” or freedom of movement would better test mesh is consistent across dramatically different views and does not have holes. As a viewer, it is also closer to the desired goal of exploring the generated scene.

-	If the paper were instead focused on the infinite-generation direction, perhaps a more appropriate set of experiments and direction would be those relating to Infinite Nature?

Generated videos are of houses with unrealistic structure, breaking the desired illusion

-	The trajectory of cameras means the video moves backward with a zig-zag type pattern. Walls tend to move correspondingly, and are typically equally at both edges of the video.

-	This results in sequences where the viewer moves down an infinitely long hallway that goes back and forth as the camera does. This feels unrealistic; I can’t imagine an indoor structure that looks like any of the videos.

-	I feel prior work (e.g. GEN-1 or VideoFusion) actually may generate houses that are in some ways more realistic, as they do not always have walls at the edge of the video frame (although the difference in movement between each frame makes the comparison difficult to judge)

Experiments feel somewhat ad-hoc. The small size and lack of consistency / justification for prompts make it harder to draw conclusions from results

-	Ablations take place on 20 videos using 10 text prompts (small set). It is also surprising to use only 10 text prompts when generating 1000 videos for the VF experiment; or 5 text prompts on 110 videos against GEN-1. Results would be much more compelling if using a different prompt for each video. I’m curious if there is a limitation I am missing that could explain this.

-	Prompts are not programmatically generated, and I did not find a justification for the selected prompts (lack of consistency; do not reflect some true data distribution). Programmatic generation of prompts would also more easily enable scaling to larger sets.

-	Prompts contain significant unique detail e.g. “indoor scene, interior, candy house, fantasy, beautiful, masterpiece, best quality” and “POV, haunted house, dark, wooden door, spider webs, skeletons” (feels as if it could be prompt engineered, doesn’t feel reflective of some true distribution).


Edit: Thanks for the rebuttal. I think the results with more significant turns are better, and showing some examples without prompt engineering is helpful.

I feel after reading the rebuttal and other reviews the paper is still a 3 - reject. I stand by my original comment the lack of systematic experiments in the paper make it hard to draw clear scientific conclusions from this work. In additon, while results are consistent across frames, the completely unrealistic nature of generated houses mean this contribution is not impressive enough for acceptance in my mind.

The reviews of this paper are very mixed. Despite my negative review, accepting it would not be the end of the world if other reviewers end up feeling very positive given (1) the generated frame quality and cross-frame consistency and (2) a few technical contributions e.g. floaters, finetuning depth.

**Questions:**

-	Motivating the intro with single-image novel view synthesis is a strange choice, and is a bit confusing, when the focus of this work is text-driven scene synthesis
-	Missing period L126
-	Slightly strange writing to see a list (L130) enumerated into subsections. It would feel more typical to change this sentence to finish with a period, and overview the subsections, or to simply make subsections as paragraphs.
-	One addition metric that could be useful is CLIP R-Precision, which is used by prior work e.g. DreamFusion. By selecting some random frames in each video and ranking CLIP distance to all text prompts, one can compute precision. This could be helpful as automated visual semantic metrics can be limited in a setting such as this.

**Limitations:**

Yes

---

> ### Author Rebuttal · Authors · 2023-08-09
>
> We appreciate the comments and feedback from the reviewer.
>
> ### Camera motion and scope
> Please refer to our global response regarding our choice of default camera trajectory. Please also refer to the attached PDF and the provided video for sample extended results, including large camera turns and circular motion. Indeed, as an inspirational example, we describe in the introduction the ability of humans to envision and expand a scene in our mind by observing only a single photo of it. We do not claim that the task is yet solved computationally but believe that our work takes a significant first step in this direction. We are the first to show how to harness the generative prior of a text-to-image model for the task of perpetual view generation and thus focus on the fundamental challenges involved in achieving a 3D-consistent video. We believe that our work will trigger future research in this direction, e.g., enabling arbitrary camera motion, as suggested by the reviewer. Finally, we would like to note that our primary goal is not to create a perfect mesh but rather to use its geometric properties to synthesize a 3D-consistent video. Thus, our default camera trajectory and evaluation metrics were selected accordingly.
>
> ### Metrics
> We indeed considered the series of InfiniteNature papers. However, their evaluation metrics are not suitable for our case. Since these methods are GAN-based models trained on specific domains (e.g., landscape videos), they use FID and other reconstruction metrics to evaluate their visual quality. These metrics are not applicable in our case since we do not have ground truth training videos or specific underlying data distribution to match (L273 in the paper).
>
>
> ### Generated content quality
> The quality of the generated content of our method mostly depends on the generative prior and the depth estimation models working together. We believe that our method of combining those priors is a major step towards text-driven perpetual view generation.
> As described in section 4.3.2 in the paper, to assess the visual quality of our outputs and to compare with the existing works, we performed an AMT user study against prior works -  VideoFusion and GEN-1. As can be seen in Table 2, our method was preferred in 96% and 69% of the cases for VideoFusion and GEN-1, respectively.
> Additionally, in the case of GEN-1, it takes the scene geometry as input, thus tackling a simpler task.
>
> ### Prompt selection
> We focused on synthesizing static indoor scenes, thus any prompt that describes such scenes can be inputted into our method. We appended a few words such as “masterpiece, best quality” to improve the visual quality and the fidelity to the text prompt, which is common practice in numerous papers  [61,62,63]. Following the reviewer’s comments, we omitted these words and re-run our method. As seen in the attached PDF and in the third row in the provided video, our method is agnostic to these specific words and works well with very simple scene descriptions. We will clarify this in the paper.
>
> Regarding evaluation, our goal was to base our quantitative evaluation on prompts that we could also qualitatively show and evaluate with a user study. Thus, we selected a set of diverse, representative prompts (see SM for the full list).
>
> Finally, as a proof of concept, we also verified that our method works well on ten prompts generated by ChatGPT when it was asked to generate prompts describing indoor scenes. We will include these results in our revised version.
>
> [61] Ben Poole, Ajay Jain, Jonathan T. Barron, Ben Mildenhall: DreamFusion: Text-to-3D using 2D Diffusion. ICLR 2023
>
> [62] Alexander Quinn Nichol, Prafulla Dhariwal, Aditya Ramesh, Pranav Shyam, Pamela Mishkin, Bob McGrew, Ilya Sutskever, Mark Chen: GLIDE: Towards Photorealistic Image Generation and Editing with Text-Guided Diffusion Models. ICML 2022
>
> [63] Aditya Ramesh, Prafulla Dhariwal, Alex Nichol, Casey Chu, Mark Chen: Hierarchical Text-Conditional Image Generation with CLIP Latents. CoRR abs/2204.06125
>
> ### CLIP R-precision
>
> Following the reviewer's suggestion, we evaluated the CLIP-R precision. We measured the accuracy with which CLIP retrieves the caption that was used to generate a frame among all the prompts used in our evaluation (10 prompts in total). Our method achieved a score of 0.92, while the naive warp-inpaint baseline scored 0.82. We will add this metric to our revised paper.

---

> > ### Comment · Reviewer_Zj8B · 2023-08-15
> > **Rebuttal Response by R4 (Zj8B)**
> >
> > Thanks for the rebuttal. I think the results with more significant turns are better, and showing some examples without prompt engineering is helpful.
> >
> > I feel after reading the rebuttal and other reviews the paper is still a 3 - reject. I stand by my original comment the lack of systematic experiments in the paper make it hard to draw clear scientific conclusions from this work. In additon, while results are consistent across frames, the completely unrealistic nature of generated houses mean this contribution is not impressive enough for acceptance in my mind.
> >
> > The reviews of this paper are very mixed. Despite my negative review, accepting it would not be the end of the world if other reviewers end up feeling very positive given (1) the generated frame quality and cross-frame consistency and (2) a few technical contributions e.g. floaters, finetuning depth.

---

> > > ### Author Response · Authors · 2023-08-16
> > >
> > > We thank the reviewer for the feedback.
> > >
> > > We respectfully disagree that our experiments are not systematic. We thoroughly evaluated the key components of our method: geometric consistency and visual quality, using various metrics as well as an extensive user-study (Table 1 and Table 2 in the paper). We demonstrated a significant boost in all metrics compared to state-of-the-art video generation and editing methods (GEN-1, VideoFusion). The size of our evaluation set is comparable to other  *test-time training/optimization* frameworks in the context of 3D/video generation, e.g. Text2Mesh [30] - 8 meshes, VOLSDF [66] - 24 scenes, NERF [31] - 20 scenes, IDR [54] - 15 scenes, Text2LIVE [3] - 7 videos.
> > >
> > >
> > > [66] Lior Yariv, Jiatao Gu, Yoni Kasten, and Yaron Lipman. Volume rendering of neural implicit surfaces. In Thirty-Fifth Conference on Neural Information Processing Systems, 2021.

---

### Official Review · Reviewer_9ntp · 2023-07-06

**Soundness:** 3 good
**Presentation:** 4 excellent
**Contribution:** 4 excellent
**Rating:** 8
**Confidence:** 4

**Summary:**

1. This paper proposes a method for test-driven consistent scene generation using a pretrained text-to-image model and depth prediction models.
2. The updating of mesh representation and fine-tuning of the decoder and depth prediction module enable continuous scene generation.

**Strengths:**

1. The continuous scene generation demonstrated in both the main text and supplementary materials proves the effectiveness of the method. The proposed method generates more continuous results than the baseline method, given the same initialization and camera trajectory.
2. It is interesting to see the fancy effects achieved by utilizing pretrained inpainting and depth prediction models.

**Weaknesses:**

1. There is a lack of analysis on cumulative errors.
2. The efficiency analysis provided in the supplementary materials could benefit from further optimization.

**Questions:**

1. The paper could provide additional analysis on cumulative errors and failure cases to further demonstrate the robustness of the proposed method.

**Limitations:**

The paper does address the limitations of the proposed method.

---

> ### Author Rebuttal · Authors · 2023-08-09
>
> We appreciate the comments and the feedback from the reviewer.
>
> ### Error accumulation analysis
>
> Following the reviewer’s comment, we measured our error accumulation for our video evaluation set. To do so, we measured: (i) visual quality by measuring the CLIP aesthetic score as a function of frame time step and (ii) geometric consistency by evaluating COLMAP on the first $n$ frames ($n$=10,20,30,40,50) and computed the corresponding SI-RMSE as a representative metric for the geometric consistency.
>
> As can be seen in the attached PDF, our method exhibits slight degeneration in visual quality, as expected and discussed in Section 5 in the paper, yet results in a significant improvement over the warp-inpaint baseline. Regarding geometry consistency, our SI-RMSE remains steady or even slightly improves with the length of the video (since COLMAP works better when more correspondences can be established).

---

### Official Review · Reviewer_ooiE · 2023-07-12

**Soundness:** 3 good
**Presentation:** 4 excellent
**Contribution:** 2 fair
**Rating:** 5
**Confidence:** 4

**Summary:**

In this paper, the authors study text-driven perpetual view synthesis with the input of text prompts describing the scene and desirable camera poses. Instead of training models on a specific domain, they propose an online scene generation method with a pretrained text-to-image model and a pretrained monocular depth prediction model. To further improve the video consistency, they utilize a shared mesh to represent the generated scene and deploy online time-time training to finetune the monocular depth prediction model for better mesh construction. The experiments demonstrate that the proposed method can generate geometrically-plausible scenes, and the proposed modules can improve the video quality quantitatively and qualitatively.

**Strengths:**

- The authors directly deploy a pretrained text-to-image model and a pretrained monocular depth prediction model as a strong indutive bias, enabling their method to work on diverse scenes in a zero-shot manner and eliminate training datasets.
- The generated videos are consistent and geometrically plausible. With the text prompts, the users can decide on the generated scenes.
- To further improve the online generated scenes, the authors deploy a unified mesh representation and adopt test-time training to finetune the depth prediction model, which are validated to improve the results in the extensive experiments.


**Weaknesses:**

- The main concern of the proposed method is that the presented results are nearly all with camera trajectory moving back. Can the proposed method work on trajectories similar to those in Text2Room? Or does some inherited property limit it to work on those camera trajectories?
- I have a question on the need for the decoder finetuning. When we want to apply the text2image diffusion model for inpainting, we can replace the existing part with the input though the decoder can not perfectly reconstruct the input. In this case, why do we need decoder finetuning?
- When watching the visual ablation study, it is actually hard to distinguish the improvement of the proposed modules. The demonstrated videos are nearly all moving backward and corridor scenarios. In this case, only the warp-inpaint baseline will work well.
- In Line 222, you mention the depth-aware splatting. Can you explain how you elaborate more on this baseline?
- Why is the mesh so blurry in Figure 7 (left)? Worldsheet also uses the mesh representation, but their boundary is not that bad. Have you tried to split the mesh into foreground and background to help the discontinuity of the scene?
- In the supplement video results, the warp-inpaint baseline is missed for visual ablation.

**Questions:**

Please answer the questions mentions in the weakness part. I will adjust my score according to the rebuttal.

**Limitations:**

The authors have properly addressed the limitation and broader impact in the main paper and supplement.

---

> ### Author Rebuttal · Authors · 2023-08-09
>
> We appreciate the comments and the feedback from the reviewer.
>
> ### Camera Motion
> Please refer to the global response section, where we elaborate more on our camera motion.
> Text2Room (concurrent work) focuses on generating a mesh rather than synthesizing a coherent video as we do. Thus, they tailor their camera poses to increase the quality of the generated mesh (e.g., covering holes). Specifically, they consider many short, segmented, and distant cameras. This choice is unsuitable for representing a coherent video captured by a single camera smoothly moving in space.
>
> ### Decoder finetuning
> As discussed in Sec. 3.4 (L156), directly combining the known content (rendered from the mesh) and the new inpainted content results in a visible seam at the inpainting border. This is caused due to imperfect encoding/decoding of the Latent Diffusion Model. By finetuning the decoder to accurately reconstruct the input content at the known pixel locations, we resolve this issue.
>
>
> ### Visual ablation study
> The warp-inpaint baseline is provided in the Ablations section in the HTML SM file, in the rightmost column under the name “naive warp-inpaint”.
>
> We agree that the differences between some of the ablation videos, when all videos are played side-by-side, may be difficult to perceive due to the fast camera motion and overall similarity of the generated content. Note that in our user study, only two videos are played simultaneously, which makes the visual comparison easier. As seen in Tables 1 and 2 in the paper, most users preferred our full method. This is further supported by the CLIP aesthetic score.
>
> ### Depth-aware splatting
> The depth-aware splatting baseline is part of the ablation study and evaluates the contribution of having a global unified representation. As an alternative way to generate geometrically consistent video without any unified representation, this baseline generates new frames using a splatting warp [33] of the previous frame content into the next frame. This method uses the depth and the camera motion from the previous known frame to define the warping field. As can be seen, our method outperforms this baseline, indicating the importance of having a unified representation.
>
> ### Outdoors scenes
> Figure 7 shows the result of our method without any special handling for outdoor scenes. Splitting the mesh into foreground and background layers, as the reviewer suggests, could help handle the scene's discontinuity. However, when we tried such an approach, we noticed that two layers are usually not enough to support challenging cases of diverse content with multiple depth discontinuities, e.g., a scene of rocks, trees, distant mountains, and cloudy skies. Furthermore, It requires handling segmentation inaccuracies. Therefore, we think that outdoor scenes require special handling and would be exciting future work.

---

> > ### Comment · Reviewer_ooiE · 2023-08-21
> >
> > Thanks for your rebuttal. The proposed method proposed an easy yet effective method for perceptual view generation. After reading other reviews and the rebuttal, I decided to keep my score as BA.

---

### Official Review · Reviewer_R7aK · 2023-07-14

**Soundness:** 4 excellent
**Presentation:** 3 good
**Contribution:** 3 good
**Rating:** 7
**Confidence:** 4

**Summary:**

This paper proposes a text-driven perpetual view generation method that synthesizes long-term changing views of a scene and its corresponding 3D mesh, given the free-vocabulary text and camera trajectories. The authors apply an off-the-shelf pre-trained text-to-image diffusion model and a pre-trained depth prediction model to achieve the baseline quality of novel-view synthesis by reprojection and inpainting, and also apply a test-time finetuning scheme to enhance the geometry consistency and textural consistency throughout the view sequence.

**Strengths:**

1. The paper is easy to follow and provides many impressive qualitative results.
2. It is promising that this method can synthesize diverse and zero-shot scenes without large-scale training data on a specific target domain.
3. The SceneScape pipeline is technically sound to generate consistent views as its novel views are grounded in the scene geometry, where the generation of novel contents and geometrical structures can be readily adapted from off-the-shelf text-to-image diffusion-based inpainting/generation models and depth prediction model.
3. By maintaining a unified 3D mesh representation and adopting a test-time finetuning technique, this method achieves even more consistent results. Such a technique is meaningful for camera pose-conditioned or other geometry-aware online scene generation pipelines.


**Weaknesses:**

I am almost satisfied with this manuscript. Here just list a few questions other than the limitations discussed in the paper, as follows:

1. The SceneScape seems not to be able to handle dynamic foreground content. If the textual descriptions tell the model that it has to generate ``dancing people'', what will SceneScape do? Moreover, as one possible downstream application, how to edit the perpetual scenes generated by SceneScape?

2. There are still quite a few strange distortions in the synthesized scenes. Can it be further improved by some instance-level priors? For example, knowing that there exists a chair, and we can use the prior knowledge of chairs to rectify the shape distortions? Maybe adding the prediction of instance segmentation maps into the SceneScape pipeline will be helpful.

**Questions:**

1. The footage provided in the supplementary is all about a walk-through towards a straight corridor. Curious about if it can handle more complex situations, e.g., turning left/right/back?

2. How many prompts are used in total in the test stage? Can you give a full list of them? It is unclear if the prompts cover various indoor scenarios and further affect the metrics.

3. It seems time-consuming to synthesize one frame according to the supplementary material. A comparison of the runtime with the reference methods would be helpful.

**Limitations:**

The authors have stated limitations and potential negative societal impact in the paper.

---

> ### Author Rebuttal · Authors · 2023-08-09
>
> We appreciate the comments and the feedback from the reviewer.
>
> ### Dynamic content generation
> By design, our method supports only static scenes since we use a shared unified mesh representation for all the frames. Thus, when providing SceneScape with a prompt that generates an object with dynamic nature, it stays fixed throughout the entire video. An example of such behavior is the video depicting a walkthrough inside a medieval forge provided in the SM. As can be seen, the fire in the furnace is static. Adapting our method to support dynamic scenes is an exciting avenue for future research.
>
>
> ### Improved geometry
> We thank the reviewer for the interesting suggestion. Indeed the quality of our results is affected by the accuracy and consistency of the depth predictions, which we improve through depth finetuning, but are still not perfect. Incorporating additional object-level shape priors can potentially further improve the estimated geometry of specific objects, yet it is a non-trivial task since object-level information needs to interact consistently with the rest of the generated scene content. Note that our method is not restricted to a specific depth prediction model, thus we can leverage any future improvement in monocular depth prediction when it becomes available.
>
> ### Camera motion
> Please refer to the global response section, where we elaborate on our camera motion.
>
> ### Text prompts
> In our ablations, we use 20 videos with a set of 10 diverse text prompts, which are provided in section 2 in the Supplementary Materials. We use a diverse set of representative prompts describing real and fantasy indoor scenes, dark and lit scenes, as well as buildings and natural structures (cave).
>
> ### Runtime w.r.t. baselines
> VideoFusion takes ~15 seconds for a 16-frame video of 256*256 resolution on a TeslaV100 GPU. Since we do not have direct access to the GEN-1 model, we are limited in assessing its runtime. To get a rough estimate, we measured their online interface runtime, which takes ~1 minute to edit a video with 25 frames. Our runtime, along with the runtime of each of our modules, are included in Section 1 of the SM. Please note that our method employs test-time optimization/training, whereas the baselines are feedforward models, thus the runtimes cannot be directly compared. We will include the runtime comparison in the revised SM version.

---

### Author Rebuttal · Authors · 2023-08-09

We would like to thank the reviewers for their thorough and valuable feedback. Please note that a link to the video file has been sent to the AC with additional results. Please see the per-reviewer responses for details.

### Camera motion
The reason we focused on backward rotational camera motion is twofold: (i) our primary objective is synthesizing long coherent videos. Thus, the camera trajectory has to follow a smooth path that complies with the generated scene, i.e., avoid crashing into walls or going out of the scene's bounds. (ii)  synthesizing the scene while zooming in / moving forward into the scene would require adding new details to existing content, i.e., would require solving the challenging task of super-resolution. Empirically, we found that backward rotational camera motion allows us to bypass these challenges without any user input or manual control while demonstrating non-trivial parallax effects, occlusion/disocclusion handling, and complex scene geometry.

Following the reviews, we expanded our results to include camera paths that depict sharp turns and backward spiral motion. As can be seen in the attached PDF and the provided video (first and second rows), our method performs well under these challenging camera paths. We will include these results and discussion in the revised paper.

We believe that a more sophisticated, scene-aware camera path may be possible to design yet out-of-scope of our paper that considers and tackles the task of text-driven perpetual view generation for the first time.

---

### Decision · Program_Chairs · 2023-09-21

**Decision:**

Accept (poster)

**Comment:**

The paper has mixed reviews. On the positive end, the paper is one of the first explorations toward the text-driven perpetual view generation using pretrained text2image model, and shows reasonably well quality and consistency. On the negative end, there are some concerns regarding camera trajectory, visual quality in some scenarios, insufficient evaluations, and inaccurate claims. I believe most of the concerns are addressed after the rebuttal. I would like to recommend the acceptance of the paper, and suggest the authors revise the manuscript accordingly.